# Effect of Relative Humidity on the Population Dynamics of the Predator *Amblyseius swirskii* and Its Prey *Carpoglyphus lactis* in the Context of Slow-Release Sachets for Use in Biological Control in Greenhouses

**DOI:** 10.3390/plants11192493

**Published:** 2022-09-23

**Authors:** Yohan Solano-Rojas, Juan R. Gallego, Manuel Gamez, Inmaculada Lopez, Patricia Castillo, Tomas Cabello

**Affiliations:** 1Research Centre for Mediterranean Intensive Agrosystems and Agrifood Biotechnology (CIAMBITAL), Agrifood Campus of International Excellence (CEIA3), University of Almeria, 04120 Cañada de S. Urbano (La), Spain; 2Department of Mathematics, University of Almería, 04120 Cañada de S. Urbano (La), Spain

**Keywords:** horticultural crops, microclimatic conditions, inoculative releases, prey–predator, mathematical model

## Abstract

*Amblyseius swirskii* is a predatory mite that is widely used for biological control in greenhouses. One way this predator is released is in a formulation in slow-release sachets. These sachets are prepared with the predatory mite, the factitious prey mite *Carpoglyphus lactis*, and a food substrate for the latter. The objective of the present study was to study the effects of microclimatic conditions in this type of formulation on the population dynamics of mites inside the sachets and on the release of predatory mites. These experiments were conducted under laboratory conditions in two trials. The ambient relative humidity affected the water content of the food substrate of the prey mite inside the sachets, with an initial value of 21.49 ± 0.42%, which was reduced to values of 4.7 ± 0.25%, 10.87 ± 1.03% and 17.27 ± 0.82% after 21 days of trials when they were exposed to low, medium and high ambient relative humidity levels, respectively. Relative humidity significantly altered the dynamics of the populations of both species inside the sachets and the exits of the predator from the sachets to the external environment.

## 1. Introduction

The predatory mite *Amblyseius swirskii* (Athias-Henriot) (Acari: Phytoseiidae) is widely used worldwide for the control of small insects and mites that cause damage to crops [1,2]. This organism is successful because some species are omnivorous and can complete their biological cycle by feeding on pollen, which allows them to establish on the crop in a preventive manner and even develop in the absence or scarcity of pests [3,4]. Information on its origin, distribution and biological characteristics in relation to its use in biological control of this species is reviewed by Calvo et al. [5].

Environmental conditions are more stable and controlled for protected crops than for other outdoor crops. However, parameters such as temperature, relative humidity (RH) and light intensity vary over time. The main factors that influence these fluctuations are climate, season, geographic location, type of crop and the crop’s plant structure [6]. The establishment of a population of predatory mites and their effectiveness in pest control will depend on the environmental conditions of the crop under field conditions [6]. Microclimatic conditions such as temperature and RH are crucial factors contributing to the survival of these phytoseiids [2] and influence the increase in their populations. Specifically, RH determines the viability of eggs and the developmental success of adults [6].

The two typical methods for the release of phytoseiid mites are direct release on the plant, through inoculative or inundative release when pest species are present, and the more recently developed novel method of slow-release sachets [7].

The slow-release sachets ensure better colonization and installation of the predator in the crop, which is key for successful biological control [8,9]. These sachets release a large number of predatory mites over a sustained period [10] over several weeks [6]. They consist of a sachet with water-repellent walls and an outlet opening. The interior contains wheat bran along with sawdust or vermiculite as a carrier medium and food for the factitious prey, which is the food source for *A. swirskii*, ensuring its development until its spontaneous exit to the crop [8,9,11,12,13]. An alternative prey used as food for *A. swirskii* in slow-release sachets is *Carpoglyphus lactis* (L.) (Acari: Carpoglyphidae) and *Tyrophagus putrescentiae* (Schrank) (Acari: Acaridae) [5,8,14].

The commercial production of predatory mites is easy if the biological requirements of these arthropods are met. However, in the agroecosystems, large populations of phytoseiids are not frequently observed, suggesting that some natural factors affect their abundance in the field crops [1]. Therefore, an understanding of the effects of the parameters involved on the life cycle is necessary to evaluate the potential growth of the population of the species [2].

RH is one of the most important microclimatic factors that influences the biological parameters of mite populations [1,2,10,15], as it is crucial for their survival [1,2]. It directly affects the net reproduction rate and the intrinsic rate of natural increase for both *A. swirskii* and *C. lactis* along with other parameters that determine the population dynamics of predators [2]. In most phytoseiid mites, the egg stage is the most sensitive to dry conditions [1,10]. Thus, an RH ≤ 70% reduces the viability of eggs [10,16]. The motile forms (larvae, nymphs and adults) of the mites are capable of restoring their water content through food, consuming free water, seeking favorable places, etc. [1]. In this sense, a high RH may alleviate extreme conditions such as extremely hot temperatures and water scarcity [1,2,15]; in contrast, low RH may limit population growth at optimal temperatures, especially if the mite uses energy to balance hydration at the expense of reproduction [1,2].

Based on the information described above, the environmental conditions inside the slow-release sachets seem to determine both the proliferation of predatory mites and prey mites. Therefore, the objectives of this work were twofold—firstly, to try to improve the evaluation of the slow-release sachet system by fitting it to mathematical models, and secondly, to evaluate the effect of RH on this biological control system and to discuss its repercussions for greenhouse crops.

## 2. Results

The results obtained in the trial are described below in four sections. Section 2.1 shows the variation in the RH of the environment (three treatments) and its effect on the humidity inside the sachets along with the water content in the substrate at the beginning and end of the trial. And, the results of the population dynamics of both mite species inside the sachets at each ambient humidity regime are provided. Subsequently, in Section 2.2, the effects of these RH levels on the exits of the predatory mite from the sachets are shown. Finally, in the fourth and closing Section 2.3, the results of the fit to the mathematical model of these mite release data are presented.

### 2.1. Trial 1: Dynamic Changes in the Populations of A. swirskii and C. lactis Inside the Sachets According to RH

Figure 1 shows the variation throughout the RH trial inside the sachets (solid lines) as a function of the RH (dotted lines). Thus, the internal RH of the sachets having a 22.5% RH in the environment decreased substantially during the first 7 days from 72.80% to 20%, at which point the internal RH of the sachet fluctuated within the range of 14–16% until the end of the trial. In turn, the RH curve inside the sachets from the 52.5% RH regime progressively decreased from 70.8% to 43.1%, remaining above 60% during the first ten days. Finally, the internal RH of the sachets exposed to 87.5% ambient RH increased from 73.1% to 85% on Day 8 and then remained within the range of 84–86.48%.

By comparison, the water content of the substrate varied from the initial value of 21.49 ± 0.42% to values of 4.78% ± 0.25%, 10.87 ± 1.03% and 17.27 ± 0.82 after 21 days of exposure to the low, medium and high RH values, respectively.

Figure 2A,B show the temporal variation in the populations of the prey mite *C. lactis* and the predatory mite *A. swirskii* (motile mites, in both cases) inside the slow-release sachets when exposed to three different relative humidity levels outside the sachets.

The statistical analyses performed using generalized linear models (GZLMs) for the data on the total number of *C. lactis* prey mites per sachet showed the high statistical significance of the model used (omnibus test: chi-square test of the likelihood ratio = 104.610; degrees of freedom d.f. = 11; *p* < 0.01). Highly significant effects of the factors analyzed were observed, namely, RH (chi-square likelihood ratio = 25.950; d.f. = 2; *p* < 0.01), trial time (days) (likelihood ratio chi-square = 96.399; d.f. = 3; *p* < 0.01) and their interaction (chi-square likelihood ratio = 21.715; d.f. = 6; *p* < 0.01), on motile factitious prey mites *C. lactis* inside the sachets.

According to the results (Figure 2A), the population of the prey mite exhibited a progressive decrease in all treatments, except for the high RH treatment (87.5%), where the population increased slightly from its initial density of 4235 ± 154.73 mites/sachet to 4352.5 ± 409.46 mites/sachet on Day 4. However, this density was not significantly different from that recorded at medium ambient RH, even until Day 7, because on Day 15, the population observed at this RH was significantly higher than that of the other treatment groups. At low ambient RH (22.5%), the *C. lactis* population remained consistent from Day 4 to Day 15, with values lower than those recorded in the other two RH groups. Finally, on Day 21, the three treatments did not differ significantly in the mite population.

Regarding the data for the motile predator mites *A. swirskii* inside the slow-release sachets, the statistical analysis using GZLM indicated the high significance of the model used (omnibus test: chi-square likelihood ratio = 69.421; d.f. = 11; *p* < 0.01). The factors analyzed, namely, ambient RH, time (day) and their interaction, exerted highly significant effects on the total number of motile mites of this species (chi-square likelihood ratio = 18.896; d.f. = 2; *p* < 0.01; chi-square likelihood ratio = 53.746; d.f. = 3; *p* < 0.01 and chi-square likelihood ratio = 20.377; d.f. = 6; *p* < 0.01; respectively).

As shown in Figure 2B, on Days 4 and 7, a significantly greater number of motile *A. swirskii mites* was observed in the sachets exposed to a medium RH of 52.5% (1867.5 ± 210.85 and 2135 ± 74.44 mites/sachet), than in sachets exposed to 87.5% RH (1587.5 ± 142.50 and 1532.5 ± 150.08 mites/sachet) and 22.5% RH (1362.5 ± 165.04 and 1065 ± 103.16 mites/sachet), respectively, while on Days 15 and 21, this value did not show significant differences between treatments. In general, the population of *A. swirskii* treated with low or high ambient RH progressively decreased from the initial value (Day 0), while at medium RH, the population peaked on Day 7.

### 2.2. Trial 2: Evaluation of the Release of A. swirskii from the Sachets, According to RH

Figure 3 shows the temporal variation in the total motile mites of the predator *A. swirskii* that hatched from each sachet after exposure to the three levels of RH. The values for treatment and day represent the number of total mites accumulated during the intervals of days between sampling (0–4, 5–7, 8–15 and 15–21, respectively).

In the statistical analysis of the previous data, which was also performed using GZLM, a highly significant effect of the model used was identified (omnibus test: chi-square likelihood ratio = 65.376; d.f. = 11; *p* < 0.01). Highly significant effects of the RH factor (chi-square likelihood ratio = 20.096; d.f. = 2; *p* < 0.01), the time factor (day) (chi-square likelihood ratio = 51.199; d.f. = 3; *p* < 0.01), and the interaction of both factors (chi-square likelihood ratio = 58.813; d.f. = 6; *p* < 0.01) were observed.

In the period of 0–4 days, the release of *A. swirskii* did not differ significantly between the three treatment groups, while in the period of 5–7 days, the number of predators that exited the sachets at low and medium RH did not differ significantly from each other. Both numbers were significantly higher than the value obtained at a low RH. The latter value corresponds to the maximum of this treatment (137 ± 21.69 mites/sachet). Subsequently, the release of predatory mites at low RH always remained significantly lower than that at the other two humidity levels. In turn, at medium and high RH, the values increased in the samples to maximum values in the interval of 8–15 days, when they reached their maximum values (262.25 ± 37.88 mites/sachet and 198.75 ± 36.24 mites/sachet, respectively). Subsequently, the exit of predators gradually decreased until the end of the trial.

### 2.3. Fit of the Mathematical Model to the Number of Predatory Mites Released from the Sachets: Rate and Period of Mite Release, According to the RH Regime

The previous data for the exit of motile mites of the predator *A. swirskii* (Trial 2) (Section 2.2) may be better interpreted by the mathematical model used. Thus, Figure 4 shows the Verhulst–Pearl logistic models fitted to the total cumulative population (motile mites) that hatched from the sachets at the three values of ambient RH evaluated and their derivative functions. These fits were highly significant (Table 1).

Although it is simpler, this mathematical model seems to better represent the actual conditions observed when establishing the exit speed of the predatory mites from the sachets (derivative functions) and their release window (period) (Figure 4).

The total number of predatory mites released (accumulative values) from the sachets at low RH was less than the value of 300 motile mites/sachet (Figure 4C). This value is the standard number for these formulations established by the International Organization for Biological Control (IOBC) [17]. In addition, the release period was noticeably short at 2–3 days. In contrast, at medium and high RH levels, the accumulative number of the total mites released was higher at approximately 500, covering periods of 10–15 days (Figure 4A,B).

## 3. Discussion

The importance of the microclimate (usually RH) in the performance of beneficial arthropods (predators and parasitoids) was established more than 80 years ago by Taylor [18] and is caused by the possible effects of stress under nonoptimal conditions on the development and survival of these entomophages (e.g., [19,20]). The survival and other biological parameters of predatory mites are particularly affected by a lack of water and food (e.g., [21]). This finding is important, especially in greenhouses, when predatory mites are released directly on the crop. However, the use of controlled releases from mite sachets may change the microclimatic conditions, especially the availability of water, rather than food, as discussed below.

According to the results obtained here, the RH (actual RH measured inside the sachets, as well as the water content of the substrate) influenced the population dynamics of *A. swirskii* and *C. lactis* inside the slow-release sachets along with the number of predators that were released. Similarly, the moisture content inside these sachets was influenced by the RH. At 22.5% RH, the predator population did not increase, and the maximum release occurred during the first week. However, higher values of ambient RH caused higher population densities of predatory mites inside the sachet and in the number of *A. swirskii* released, although not proportionally, since these values were higher at 52.5% RH than at 87.5% RH.

The RH inside the sachets decreased during the first seven days of exposure to low RH, coinciding with the decrease in the populations of the predatory mite and its factitious prey. Subsequently, both populations remained stable (Figure 1). This finding may be because environments with low RH influence the microclimate inside the sachets, causing dehydration of the carrier substrate and/or reducing the survival of mite populations, as previously indicated [1,12,22]. At the end of trial 1, the humidity values inside the sachets subjected to this 22.5% RH level were very low, leaving the substrate practically dry (substrate water content, initial = 21.49 ± 0.42% and final = 4.78 ± 0.25%), while those that remained at a high ambient RH had lost very little water in the substrate (substrate water content, initial = 21.49 ± 0.42% and final = 17.27 ± 0.82%) at the end of the trial. In contrast, fungi induced a decrease in the general fitness of the phytoseiid mite [1]. In turn, the sachets subjected to 52.5% RH recorded values higher than 40% internal RH at the end of the experiment without the appearance of fungi. In addition, with this regime of medium RH, the humidity of the sachet remained above 60% until Day 10, which favored better biological parameters of both species and the release of predators (the water content of the substrate was reduced by half at the end of the trial). In summary, the increase in RH inside the controlled-release sachets improves water retention in the carrier media, resulting in an optimal microclimate for the biological development of mites, as previously indicated [6,10,23].

In mites, the egg is the stage of development most sensitive to low RH, since eggs cannot feed or move to obtain water from prey or the environment [1,2,16,21,24,25], whereas motile mites tend to lose water by excretion or diffusion at low RH [24,25,26]. Therefore, at 22.5% RH, decreases in both the hatching and survival of the immature stages of the predatory mite and its factitious prey were observed, coinciding with the results for *A. swirskii* reported by San et al. [2], who showed that the eggs of this species do not hatch at 33% RH. Our findings are also consistent with the results reported by Midthassel [22], who indicated that the mortality of the motile stages increases in the range of 10 to 20% RH. Thus, Ferrero et al. [16] indicated that *A. swirskii* has a hygro-preference for oviposition and/or hatching of eggs such that the fecundity and/or hatching of larvae are reduced with the decrease in RH, while the duration of its motile stages is extended [2]. Thus, researchers concluded that the reduction in the population of this mite occurred as a result of the desiccation that caused the low RH near the eggs, larvae and protonymphs of the predator, and the immature stages of *C. lactis*, which induced eventual cannibalism in the females of *A. swirskii*, which feed on conspecific protonymphs due to the scarcity of prey, causing a decrease in longevity and fecundity [27]. On the other hand, the immature stages of *C. lactis* were affected by exposure to this low RH of 22.5% because the eggs of this species need a minimum value of 58.5% RH to hatch [28].

The population growth of *A. swirskii* inside the sachets exposed to medium RH (52.5%) differs from that indicated by other authors, who found that an RH less than 50–60% limits the development of predatory mites [24,25,29]. Although the population of *C. lactis* did not experience growth, it remained higher than that of the predatory mite due to the predator–prey relationship that characterizes commercial slow-release sachets. On the other hand, as mentioned above, the hatch rate of *C. lactis* eggs at 58.5% RH is 95%; however, the larvae do not reach the adult stage at this humidity level [28]. This result would explain why the treatment at medium RH did not increase the prey population but did increase oviposition, with the eggs of this prey mite maintaining and supplying the necessary moisture to the predatory mite *A. swirskii*, as cited by Midthassel et al. [13]. The increase in populations of the predatory mite with this medium RH occurred during the first 10 days, coinciding with a moisture content inside the sachets > 60%, as already indicated. In addition, it is also consistent with the fact that the hatching of the eggs from the biofactory and/or deposited inside the sachet occurs, which will reach the adult stage in approximately 7 days and subsequently consumes prey at 53% RH during this period, producing this increase in the population of *A. swirskii*.

In general, the increase in relative humidity induces the growth in mite populations and increases their survival [2,30,31]. However, the results from the present study indicate that the growth in *A. swirskii* populations inside the sachets exposed to 87.5% RH was higher than that obtained at low humidity but did not exceed the treatment with medium humidity. The slight increase in growth in the populations of *A. swirskii* and *C. lactis* during the first four days at high ambient RH might be promoted by the increase in the internal humidity of the sachets, thus allowing the population to reach the adult stage, which was observed for *A. swirskii* at 6 days when feeding on *C. lactis* at an RH of 70% [14,32] and at 8 or 9 days for *C. lactis* when the RH was 84% [28]. After the seventh day at high RH, a decrease in the population of *A. swirskii* was observed, which corresponded to the decrease in available prey mites. This fact agrees with the study by Okamoto [28], who states that only between 49.1% and 66% of *C. lactis* larvae reach the adult stage in conditions of 84 and 94% RH, while the remaining larvae will be consumed by the predator, which shows a great preference for the youngest stages of its prey [13,33,34]. Thus, a lower availability of prey together with a decrease in the feeding of the predatory mite associated with the conditions of ambient RH [35] influences the functional response of *A. swirskii* [13], causing a reduction in the population.

The release of *A. swirskii* from the sachets also varied as a function of the RH regime to which they were exposed (Figure 4A–C). The shortest release interval (2–3 days) was recorded at low RH, a treatment at which the maximum release rate of *A. swirskii* was reached faster (Day 7) than in the other treatments (Day 15). This result is consistent with the study by Midthassel et al. [6], in which they indicated that *A. swirskii* reached its maximum release at 60% RH on Day 12, faster than with an RH of 75% (Day 17). Additionally, the results reported by Midthassel [22] seem to confirm our results by indicating that the greatest exit of *A. swirskii* from sachets occurs at 10% RH compared to others exposed to 20 or 30% RH. After this maximum release value, and particularly beginning on Day 7 (Figure 4C), relatively few predators exited from the sachets, suggesting that this behavior might be motivated by the low population density of *C. lactis* or a decrease in the prey population caused by the increase in predation of *A. swirskii* due to low humidity conditions, as indicated by Mori and Chant [35] for *Phytoseiulus persimilis* (Athias-Henriot) and by Doker et al. [36] for *Neoseiulus californicus* (McGregor), in both cases at low RH.

The lowest total number of predatory mites that hatched from the sachets was recorded at low RH (22.5%), <300, the standard value for formulations established by the IOBC [17]. In contrast, at medium and high RH, the total number of releases was higher at approximately 500. The treatment at medium RH (52.5%) resulted in the release of the highest accumulative number of *A. swirskii*. Maximum release was recorded one week after the increase in its population inside the sachets (Day 15). Therefore, the production capacity of these rearing formulations was confirmed under these medium humidity conditions, where the presence of the prey *C. lactis* seems to dampen the effect of the ambient RH on the value of the internal RH in the sachet and the water content of the substrate. Shimoda et al. [10] indicated that the water supply under low humidity conditions promotes the release of *A. swirskii* from sachets exposed to 31.5–36.0% or 69.0–81.6% RH. Similarly, San et al. [2] indicated that *A. swirskii* shows a reproduction rate at 53% RH with a water supply, which is like that obtained at 92% RH with or without an extra water supply.

At high RH, the release of *A. swirskii* during the first ten days was lower than that recorded at 22.5% RH; its release profile was also lower than that recorded at 52.5% RH. These differences might be explained by the rapid exit of *A. swirskii* under low humidity conditions and by the population dynamics of the predator and prey at medium humidity, as previously described. Several authors have noted that the release of *A. swirskii* and other phytoseiid increases with the increase in RH, and they attribute this effect to the adequate development and reproduction of the predator inside the sachets [6,10]. However, the predator–prey relationship inside the sachets determines the release of predators, since the predation rate is reduced if this ratio is high [13]. A potential explanation for this finding is that phytoseiids decrease the consumption of prey when they have high population densities [13,37] and/or in environments with high humidity [35], generating a crowded environment inside the sachets that accelerates the exit of the predator [22].

RH is one of the factors with the greatest effect on the population dynamics of predatory mites and their prey inside the breeding sachets [2,22,24]. Hence, some studies show that the increase in RH in the environment promotes a greater release of predatory mites from slow-release sachets [6,10,23]. However, our results suggest that the fluctuation in the microclimate inside these confined spaces differently affects the species of mites coexisting inside them, which have different biological and/or reproductive responses to the variation in humidity. Notably, the complex dynamics of the predator–prey system inside sachets depend on the interaction of moisture with other variables, such as the quantity and quality of the prey food, the moisture retention capacity of the substrate, the spatial complexity of the substrate, the interference of congeners [12,13,34], the porosity of the paper and other materials that constitute the walls of the sachet, the development of fungi in the carrier substrate, and the location of the sachets in the field [38,39].

Greenhouses in the Mediterranean area do not normally have heating and/or cooling systems; nonetheless, a need to meet market demands with products throughout the year exists for these greenhouses. Thus, growers must extend the crop cycles toward summer seasons, when temperatures reached inside the greenhouses are well above the optimal temperatures (T > 35 °C) and ambient RH is very low with a high differential vapor pressure (DVP > 3 kPa) [40]. Based on the results we obtained, we must note that the effectiveness of the mites formulated in release sachets is potentially affected by the microclimate of these greenhouses. This effect might be mitigated with possible water misting when the RH conditions of the greenhouses are low. This solution should be evaluated under commercial greenhouse conditions.

## 4. Materials and Methods

### 4.1. Biological Material

SWIRScontrol^©^ slow-release sachets (Agrobio S.L., La Mojonera, Almeria, Spain) with a net weight of 4.83 ± 0.12 g were used and contained *A. swirskii* and *C. lactis* mites in all life stages, wheat bran as the main dispersal element of the prey (*C. lactis*) and other food substances for the latter. The material was used within 24 h of receipt according to the handling instructions provided by the company to avoid a decrease in quality.

### 4.2. Trials Conditions

Two trials were conducted in climatic chambers at 25 ± 1 °C, with 52.5 ± 2.5% relative humidity and a light:dark (L:D) photoperiod of 16:8 h. The treatments consisted of exposing the sachets to three levels of RH: high (87.5 ± 2.5%), medium (52.5 ± 2.5%) and low (22.5 ± 2.5%). These treatments were performed in sealed plastic containers with a 4.6 L capacity, in which silica gel (SiO_2_) was introduced to adjust the low and medium RH; in turn, a saturated solution of potassium chloride (KCl) was used to achieve high RH [41].

The RH levels in each treatment group were recorded daily (every 15 min) with thermohygrometers placed inside the containers (data logger, model EBI 20-TH, Ebro^®^ Xylem Analytics Germany Sales GmbH and Co. KG Ebro, Ingolstadt, Germany). The RH was also monitored inside the sachets exposed to the three regimes using chromatic plates (chromotropic trap, medium size, Agrobio S.L., La Mojonera, Almeria, Spain) and humidity sensors provided with free software connected to configured DHT-22 humidity sensors (Sensor-Th, HAIGU) (Arduino^®^, Somerville, MA, US) to record the RH every 10 min (Arduino^®^ ArduinoHome) [42]. These sensors were introduced inside the sachets. All the recorded data were stored in an SD memory card for subsequent analysis. The final water content of the substrate inside the sachets was determined by oven-drying at 100 °C until the dry weight was constant [43].

### 4.3. Trial 1: Evaluation of the Populations of A. swirskii and C. lactis Inside the Sachets

The study of the population dynamics of the predatory mite and the prey mite inside the sachet exposed to the three levels of RH was performed by placing sachets inside the plastic containers and, in turn, the silica gel or the KCl solution according to the corresponding treatment. The numbers of motile mites (larvae, nymphs and adults) of *A. swirskii* and *C. lactis* inside the breeding sachets were counted at the beginning of the trial (Day 0). Samples were collected at 4, 7, 15 and 21 days at each RH level. The treatments were replicated four times, and one sachet was needed for each day of recording and repetition, for a total of 16 sachets per treatment. The extraction of the motile stages of *A. swirskii* and *C. lactis* from the interior of the sachets to count the populations was performed using the hexane flotation extraction method described by Gallego et al. [44].

### 4.4. Evaluation of the Release of A. swirskii from the Sachets

The exit of the predatory mites from the slow-release sachets was determined using the standard methodology of biological control companies [44], which consists of installing yellow chromatic plates to capture the specimens. In this experiment, each sachet was placed at the upper end of an 8 cm long wooden rod. This rod was then inserted into the center of a yellow plate with entomological glue (20 × 20 cm^2^), and a 3 × 2 cm^2^ fragment of dark blue shiny wrapping paper was placed in the center of this plate to achieve good dispersion of the mites and subsequent immobilization in the plate. Finally, the trap with the sachet installed was placed on an expanded polystyrene (EPS) support. The chromatic plates were replaced at 4, 7, 15 and 21 days after exposure to the three treatments. Immediately after completing the aforementioned sampling procedures, the number of *A. swirskii* motile mites present on the yellow plates and the dark shiny wrapping paper were counted using a stereo microscope. After obtaining the counts, the same sachet was placed on a new yellow plate with its respective sheet of dark shiny wrapping paper until the next sampling procedure.

### 4.5. Statistical Analysis

In trial 1, the values corresponding to the total number of motile mites (larvae, nymphs and adults) of the predator *A. swirskii* and of the prey mite *C. lactis* inside the sachets and those that exited from them were subjected to statistical analysis using GZLM and the gamma function and the logarithm link function as the distribution of the dependent variable. Two factors were considered: RH (at three levels) and time (at four levels), and their interaction. The mean values were compared in pairs using the Wald test. For this analysis, the statistical software IBM SPSS version 26 [45] was used. In trial 2, only the values corresponding to the motile mites of the predator *A. swirskii* that hatched from the slow-release sachets were analyzed; the same statistical procedure mentioned above was used for this analysis.

### 4.6. Mathematical Model Biological Material

The cumulative values of the total number of motile mites (larvae, nymphs and adults) of the predator that hatched from the sachets in each ambient RH treatment group over time were fit to a logistic-type or Verhulst–Pearl mathematical model using the following equation [46]:(1)$$x=\frac{K}{1+\left(\frac{K}{x_{0}}-1\right)\cdot e^{-r_{m}\cdot t}}$$
(2)$$\frac{dx}{dt}=r_{m}\cdot x\cdot \left(1-\frac{x}{K}\right)$$
where *x*_0_ is the initial population of the mite (*A. swirskii*); *x* = number of accumulative mites hatching from the sachet; *r_m_* = intrinsic rate of population growth; *K* = maximum capacity of the system; and *t* = time (in days). The models were fitted using TableCurve 2D software version 5.0 [47].

## 5. Conclusions

(1) RH affects the population dynamics of the predatory mite *A. swirskii* and the factitious prey mite *C. lactis* when they are formulated in slow-release sachets.

(2) These effects of ambient RH alter the release of the predatory mite from the sachet to the external environment.

(3) The low ambient RH (22.5%) decreases populations of both the factitious prey and the predatory mites inside the sachet, which are significantly lower than the medium (52.5%) and high (87.5%) humidity levels.

(4) Verhulst–Pearl logistic models show the release dynamics of the predatory mite *A. swirskii* in the context of slow-release sachets.

(5) These models show that at low RH (22.5%), the total number of predatory mites released is <300 mites per sachet, with this value representing the standard for this type of formulation established by the IOBC. In addition, the release period of these mites is noticeably short at 2–3 days.

(6) In contrast, the maximum release of the predatory mite was achieved at an intermediate RH (52.5%) with total values of approximately 500 mites released per sachet during an interval of 15 days.

(7) Finally, at high RH (87.5%), the values were slightly lower than the values obtained at intermediate RH (52.5%).

## Figures and Tables

**Figure 1 plants-11-02493-f001:**
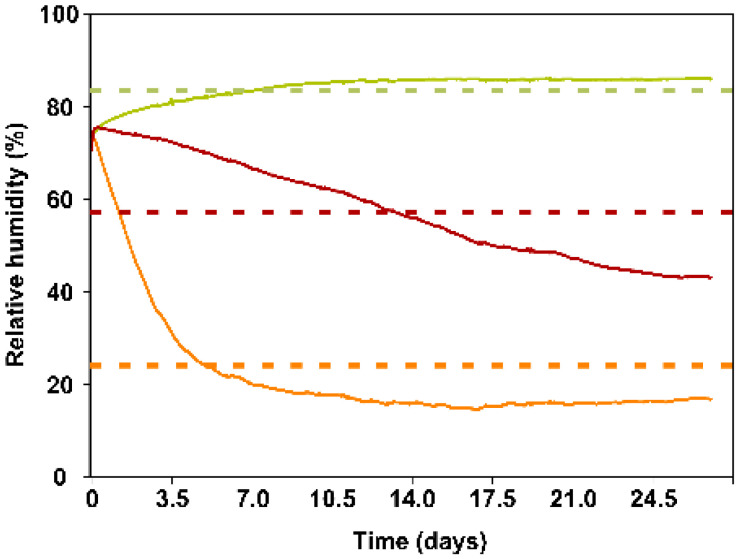
Temporal variation in the RH inside the slow-release mite sachets (solid lines) when they were maintained at three values of ambient RH, namely, low (22.5 ± 2.5%) (orange), medium (52.5 ± 2.5%) (red) and high (87.5 ± 2.5%) (green) (dotted lines), and at a constant temperature (25 ± 1 °C) under laboratory conditions.

**Figure 2 plants-11-02493-f002:**
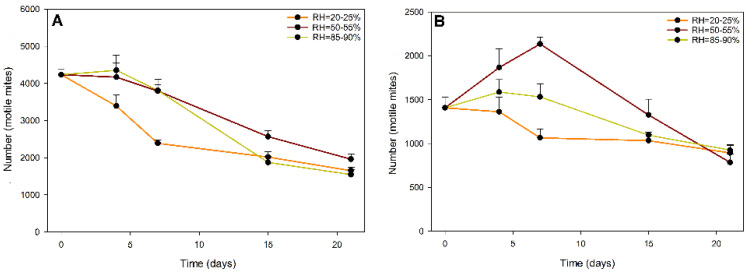
Total numbers of motile mites/sachet (means) of the prey mite *Carpoglyphus lactis* (**A**) and the predatory mite *Amblyseius swirskii* (**B**) inside slow-release sachets subjected to three levels of RH, namely, low (22.5%), medium (52.5%) and high (87.5%), in laboratory trials under controlled conditions (temperature = 25 ± 1 °C) (whiskers show the SE values).

**Figure 3 plants-11-02493-f003:**
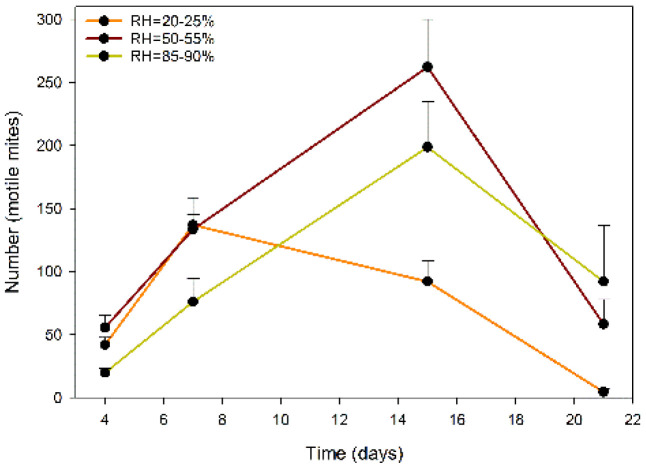
Total numbers of motile mites/sachet (means) of the predatory mite *Amblyseius swirskii* (immature motile and adult) that hatched from the slow-release sachets in periods of 0–4, 5–7, 8–15 and 15–21 days at the following RH levels in a laboratory trial conducted under controlled conditions (temperature = 25 ± 1 °C): low (22.5 ± 2.5%), medium (52.5 ± 2.5%) and high (87.5 ± 2.5%) (whiskers show the SE values).

**Figure 4 plants-11-02493-f004:**
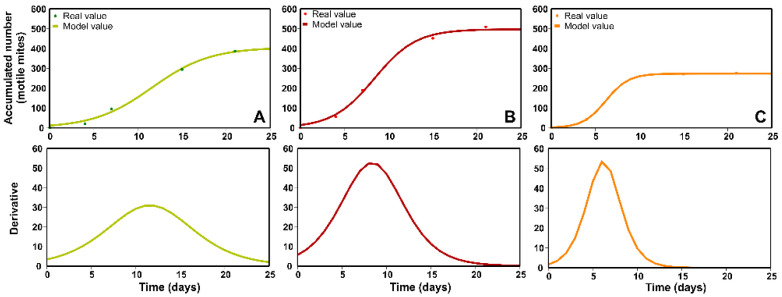
Mathematical model fits (upper panels) to the temporal variation in the total cumulative number of motile mites of the predator *Amblyseius swirskii* that hatched from the slow-release sachets at the following three humidity regimes under laboratory conditions (constant temperature = 25 ± 1 °C) and their derivative functions (lower panels): (**A**) high (87.5 ± 2.5%), (**B**) medium (52.5 ± 2.5%) and (**C**) low RH (22.5 ± 2.5%). The derivative describes the non-cumulative population size.

**Table 1 plants-11-02493-t001:** Parameters and statistical significance for Verhulst–Pearl logistic functions fitted to the total cumulative number of predatory mites *Amblyseius swirskii* (motile mites) that hatched from the slow-release sachets in the ambient RH treatment groups (22.5, 52.5 and 87.5%) under laboratory conditions (at a constant temperature = 25 ± 1 °C).

RH(%)	Fitting Parameters			Statistical Parameters		
	*K*	*x* _0_	*r*	d.f.	*R^2^*adj.	*p*
22.5	273.378	2.096	0.7857	3	0.9991	<0.01
52.5	497.206	13.805	0.4243	3	0.9815	<0.01
87.5	405.467	11.250	0.3059	3	0.9769	<0.01

## Data Availability

The data of the work will be available in the Research Data Repository of the University of Almeria, Spain: URI: http://hdl.handle.net/10835/13981 (accessed on 21 September 2022).

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
