# Peer review of "Effect of Relative Humidity on the Population Dynamics of the Predator Amblyseius swirskii and Its Prey Carpoglyphus lactis in the Context of Slow-Release Sachets for Use in Biological Control in Greenhouses"

_plants, 2022, doi:10.3390/plants11192493_

Round 1

Reviewer 1 Report

Usually, two typical methods are used for the control application of phytoseiid mites: inoculative or inundative direct release on the plant when pest species are present. The paper is interesting because, as indicated, it refers to a novel form of application of biological control, by means of small bio factories that are slow-release sachets of the predatory mite. In addition, and most importantly, it relates its mode of action to the microclimate, in this case the ambient relative humidity, as well as the water content of the sachets (actually: substrate). This is well related to its possible effects with the microclimatic conditions that can occur in greenhouse crops.

         Particularly interesting is the use, for the first time, of logistic models to explain and try to relate the ambient relative humidity values to two parameters that represent the effectiveness of the slow-release sachets: time period of predatory mite exit and the speed of them.

Minor comments to the authors:

 -Page 2, lines 65 and 66: Write “A. swirskii” and “C. lactis” in italics.

-Page 5, at the end of the legend of Figure 1, it is worth adding that “The derivative describes the non-cumulative population size”.

-Page 5, Figures 2a) and 3: Change “HR” of legend by “RH”.

-Page 5, Figure 2b): insert the same legend of Figure 2a).

-Page 5, Figure 4: From Figures 1 to 3 green color means “high RH” and orange color “low RH”, however, in Figure 4 the color code is reversed. Perhaps change the color of Figure 4a) by orange and that of Figure 4c) by green, to keep this color code. This change would not be necessary, it is only a suggestion.

-Page 6, line 244: [24, 26, 27] appear as references, reference 25 appears later (page 7, line 277).

-Page 8, line 353: Delete one dot.

-Page 10, line 417: Correct the title of Section 4.6 (Mathematical “modelBiological” material).

Summing up, with the above corrections, I suggest to accept the manuscript for publication in Plants, after minor revision.

Author Response

Reviewer #1
The authors gratefully acknowledge the comments on the first draft made by Reviewer #1, which have greatly enriched this subsequent draft.
Major comments to the Author:
Usually, two typical methods are used for the control application of phytoseiid mites: inoculative or inundative direct release on the plant when pest species are present. The paper is interesting because, as indicated, it refers to a novel form of application of biological control, by means of small bio factories that are slow-release sachets of the predatory mite. In addition, and most importantly, it relates its mode of action to the microclimate, in this case the ambient relative humidity, as well as the water content of the sachets (actually: substrate). This is well related tots possible effects with the microclimatic conditions that can occur in greenhouse crops.
Particularly interesting is the use, for the first time, of logistic models to explain and try to relate the ambient relative humidity values to two parameters that represent the effectiveness of the slow-release sachets: time period of predatory mite exit and the speed of them.
Minor comments to the authors:
-Page 2, lines 65 and 66: Write “A. swirskii” and “C. lactis” in italics. DONE.
-Page 5, at the end of the legend of Figure 1, it is worth adding that “The derivative describes the non-cumulative population size”. DONE. Excuse me, I think you mean to indicate Figure 4.
-Page 5, Figures 2a) and 3: Change “HR” of legend by “RH”. DONE.
-Page 5, Figure 2b): insert the same legend of Figure 2a). DONE.
-Page 5, Figure 4: From Figures 1 to 3 green color means “high RH” and orange color “low RH”, however, in Figure 4 the color code is reversed. Perhaps change the color of Figure 4a) by orange and that of Figure 4c) by green, to keep this color code. This change would not be necessary, it is only a suggestion. DONE. Thank you very much for the suggestion. The colors are OK, but the letters were wrong.
-Page 6, line 244: [24, 26, 27] appear as references, reference 25 appears later (page 7, line277). DONE.
-Page 8, line 353: Delete one dot. DONE.
-Page 10, line 417: Correct the title of Section 4.6 (Mathematical “modelBiological” material). DONE.

Reviewer 2 Report

In the reviewed paper, the authors aimed to investigate the effect of humidity on the population dynamics of a predatory mite A. swirskii and its prey C. lactis within sachets. The obtained results are interesting, however they are presented in suboptimal way; because of this, in my opinion the MS needs careful reconsideration and moderate revision.

The goals of the study are presented incompletely.

Some text from Discussion should be transferred to Introduction in order to help readers to understand better the study design and the main findings of the study.

The section Results needs reconsideration, it is hard to read it in its current form.

The Discussion is TOO long and poorly structured. It could be shorten 30-40%. I suggest dividing the Discussion into several subsections and provide heading for them.

Some statements that are listed in the section Conclusion needs revisions or even removal.

Finally, in the end of the paper, the authors are recommended to provide any distinct recommendations that could be used in practice based on their experimental study and explain directly what application their study could have in agriculture.

Some additional remarks are below

33-35: This organism is successful because some species are omnivorous and can

complete their biological cycle by feeding on pollen, which allows them to establish on

the crop in a preventive manner and even develop in the absence or scarcity of pests [3,4].

IS THIS THE ONLY REASON FOR THE SUCCESS? OMNIVORY IS JUST A FEATURE OF THIS SPECIES. THIS SENTENCE NEEDS REWORDING

COULD YOU PROVIDE SOME DATA ON THE NATURAL DISTRIBUTION OF THIS PREDATOR MITE SPECIES IN THE INTRODUCTION? IS IT COSMOPOLITAN, WIDELY DISTRIBUTED IN SOME BIOGEOGRAPHIC AREAS, LOCAL… ETC

41 Microclimatic conditions such as temperature and RH are crucial factors contributing to the survival of these…

DO YOU MEAN “IN NATURAL CONDITIONS” OR “IN A LAB” OR BOTH? NOT CLEAR

41-43: IT LOOKS LIKE IN THESE LINES YOU REFER TO THE SAME RESULT (BASED ON LITERATURE) WHICH YOU PRESENT AS NEW IN YOUR RESEARCH. THIS NEEDS CLARIFICATION.

58-59: in the field, large populations of phytoseiids are not frequently observed, suggesting that some factors affect their abundance in the natural environment

THE STATEMENT THAT “some factors affect their abundance in the natural environment” IS APPLICABLE FOR ANY ORGANISM, SO IT IS VERY UNCERTAIN HERE.

PHYTOSEIDS ARE FAST MOVING (SO THERE MAY BE 1-2 MITES PER LEAF BUT THOUTHANDS PER 1 TREE) AND USUALLY LAYING LESS EGGS IN COMPARISON TO PHYTOPHAGOUS MITES, THEY ARE JUST DIFFERENT. THE TEXT IN LINES 58-59 NEEDS REVISION.

65, 66: A. swirskii and C. lactis along with other parameters that determine the population dynamics of

SPECIES NAMES SHOULD BE IN ITALIC

CAPTION Figure 3. Total numbers of motile mites/sachet (means ± SE) of the predatory mite Amblyseius swirskii [ITALIC]

76: Therefore, the objective of this study was to evaluate the effect of RH on this biological control system and to discuss its repercussions on greenhouse crops.

AFTER READING THE MS IT SEEMS THAT THERE IS ONE MORE IMPORTANT GOAL OF THE STUDY. YOU LOOKED FOR THE MODEL WHICH FITS THE DATA THE BEST AND TEST THE VERHULST-PEARL LOGISTIC MODELS. THIS ASPECT OF THE STUDY SHOULD BE BETTER EXPLAINED AND PRESENTED

159: The previous data for the exit of motile mites of the predator A. swirskii (Trial 2) (Section 2.2) may be better interpreted by the mathematical model used.

THIS SENTENCE NEED BETTER EXPLANATION. “BETTER” than WHAT? WHICH MODEL WERE COMPARED? WHY EXACTLY DID YOU DECIDED TO TEST VERHULST-PEARL LOGISTIC MODELS?

209: According to the results obtained here, the RH influenced the population dynamics of A. swirskii and C. lactis inside the slow-release sachets along with the number of predators that were released.

CONSIDERING YOUR STATEMENT IN LINES 41-43 THIS STATMENT SEEMS OBVIOUS. COULD YOU EMPHASIZE WHAT EXACTLY NEW (CONCERNING RH AND MITES) WAS DONE IN YOUR STUDY?

216-217: The internal C. lactis population did not increase in any humidity regime and decreased gradually, but the decrease was faster at low RH than at medium or high humidity.

COULD YOU PROVIDE ANY BIOLOGICAL EXPLANATION FOR THIS FINDING IN THE DISCUSSION? ACCORDING TO THE FIG.2B IT LOOKS LIKE C.LACTIS REPRODUCE FASTER UNDER MEDIUM RH DURING THE FIRST WEEK BUT LATER… (THIS NEEDS SOME DESCRIPTION/EXPLANATION). ALTHOUGH YOU GIVE SOME EXPLANATION IN THE FURTHER TEXT, IT IS TOO REMOTE FROM THIS 216-217 SENTENCE.

231-234: In addition, with this regime of me dium RH, the humidity of the sachet remained above 60% until Day 10, which favored better biological parameters of both species and the release of predators (the water content of the substrate was reduced by half at the end of the trial), as will be described in more detail below.

“as will be described in more detail below” THIS SENTENCE NEEDS REVISING.

241-246: Therefore, at 22.5% RH, decreases in both the hatching and survival of the immature

stages of the predatory mite and its factitious prey were observed, coinciding with the results for A. swirskii reported by San et al. [2], who showed that the eggs of this species do not hatch at 33% RH. Our findings are also consistent with the results reported by Midthassel [22], who indicated that the mortality of the motile stages increases in the range of 10 to 20% RH.

THIS MATERIAL SHOULD BE TRANSFERRED TO INTRODUCTION. IT IS IMPORTANT TO UNDERSTAND WHY EXACTLY DID YOU APPLIED 22.5% RH AS ONE OF THE TESTED RH VALUES.

4.6. Mathematical modelBiological material

REVISE THE HEADING

426-427: (3) The low ambient RH (22.5%) decreases populations of both the factitious prey and the predatory mites inside the sachet, which are significantly lower than the medium  (52.5%) and high (87.5%) humidity levels.

WHAT EXACTLY DO YOU MEAN SAYING THIS? IT IS CLEAR THAT 22.5% IS LESS THAN 52.5% AND LESS THAN 87.5%. BUT WHY DO YOU GIVE THIS AS A CONCLUSION?

429: 4) Verhulst-Pearl logistic models show the release dynamics of the predatory mite A. swirskii in the context of slow-release sachets.

WHAT EXACTLY DO YOU MEAN SAYING THIS? DO YOU MEAN THAT THIS MODEL DECRIBES THE DATA THE BEST? OR THAT THIS MODEL FITS THE DATA THE BEST? NOT CLEAR

431: (5) These models show that at low RH (22.5%), the total number of predatory mites released is < 300 mites per sachet…

THIS CONCLUSION NEEDS REWORDING. MAY BE YOU COULD DIRECTLY SAY THAT “the total number of predatory mites released at low RH (…%) does not fit standard of 300 …”

438 (7) Finally, at high RH (87.5%), the values were SLIGHTLY lower than the values obtained at intermediate RH (52.5%).

THE WORD “SLIGHTLY” IS UNCERTAIN AND LOOKS INADEQUATE FOR CONCLUSIONS

440-442: (8) In the microclimate of greenhouses, particularly RH, plant growth of spring-summer crops and at the beginning of the growing season is lower, which may lead to ambient RH conditions that influence the actions of predatory mites, regardless of their release system, but including notably those formulated in slow-release sachets.

THIS IS NOT A CONCLUSION, THISA IS A HYPOTHESIS. THIS WOULD BE BETTER TO MENTION IN THE DISCUSSION NA DREMOVE FROM CONCLUSIONS

Author Response

Reviewer #2
The authors gratefully acknowledge the comments on the first draft made by Reviewer #2, which have greatly enriched this subsequent draft.
Major comments to the Author:
In the reviewed paper, the authors aimed to investigate the effect of humidity on the population dynamics of a predatory mite A. swirskii and its prey C. lactis within sachets. The obtained results are interesting, however they are presented in suboptimal way; because of this, in my opinion the MS needs careful reconsideration and moderate revision.
The goals of the study are presented incompletely.
Some text from Discussion should be transferred to Introduction in order to help readers to understand better the study design and the main findings of the study.
The section Results needs reconsideration, it is hard to read it in its current form.
The Discussion is TOO long and poorly structured. It could be shorten 30-40%. I suggest dividing the Discussion into several subsections and provide heading for them.
Some statements that are listed in the section Conclusion needs revisions or even removal.
Finally, in the end of the paper, the authors are recommended to provide any distinct recommendations that could be used in practice based on their experimental study and explain directly what application their study could have in agriculture.
Some additional remarks:
33-35: This organism is successful because some species are omnivorous and can
complete their biological cycle by feeding on pollen, which allows them to establish on
the crop in a preventive manner and even develop in the absence or scarcity of pests [3,4].
IS THIS THE ONLY REASON FOR THE SUCCESS? OMNIVORY IS JUST A FEATURE OF THIS SPECIES. THIS SENTENCE NEEDS REWORDING COULD YOU PROVIDE SOME DATA ON THE NATURAL DISTRIBUTION OF THIS PREDATOR MITE SPECIES IN THE INTRODUCTION? IS IT COSMOPOLITAN, WIDELY DISTRIBUTED INSOME BIOGEOGRAPHIC AREAS, LOCAL… ETC DONE: These points have been revised by Calvo et al. (2015), old [13]; a new paragraph has been added to the Introduction indicating it new [5].
41: Microclimatic conditions such as temperature and RH are crucial factors contributing to the survival of these…
DO YOU MEAN “IN NATURAL CONDITIONS” OR “IN A LAB” OR BOTH? NOT CLEAR
DONE. Added: “under field conditions”.
41-43: IT LOOKS LIKE IN THESE LINES YOU REFER TO THE SAME RESULT (BASED ON LITERATURE) WHICH YOU PRESENT AS NEW IN YOUR RESEARCH. THIS NEEDS CLARIFICATION.
ANSWER: it is detailed at the end of the Introduction (lines 76-77).
58-59: in the field, large populations of phytoseiids are not frequently observed, suggesting that some factors affect their abundance in the natural environment
THE STATEMENT THAT “some factors affect their abundance in the natural environment” IS APPLICABLE FOR ANY ORGANISM, SO IT IS VERY UNCERTAIN HERE.
PHYTOSEIDS ARE FAST MOVING (SO THERE MAY BE 1-2 MITES PER LEAF BUTTHOUTHANDS PER 1 TREE) AND USUALLY LAYING LESS EGGS IN COMPARISON TOPHYTOPHAGOUS MITES, THEY ARE JUST DIFFERENT. THE TEXT IN LINES 58-59 NEEDS REVISION. ANSWER: True, thank you for your indication. But, we are talking about agroecosystems, partly an "ecosystem" and partly 1
more a “general system". The former depends only on "natural" factors, the latter is influenced by other “non-natural factors" (e.g., farmer). In this sense, entomophagous densities in a "natural environment" are normally low; on the contrary, in agroecosystems, such densities are generally higher, because they are modified ecosystems. Thus, e.g., the migration of Phytoseids from weeds to fruit crops, e.g., citrus, is well documented. This is what is called "natural control" in agroecosystems, which is usually greater than "natural control" in natural ecosystems. DONE: Paragraph has been clarified.
65-66: A. swirskii and C. lactis along with other parameters that determine the population dynamics of SPECIES NAMES SHOULD BE IN ITALIC. DONE
CAPTION Figure 3. Total numbers of motile mites/sachet (means ± SE) of the predatory mite Amblyseius swirskii [ITALIC]. DONE
76: Therefore, the objective of this study was to evaluate the effect of RH on this biological control system and to discuss its repercussions on greenhouse crops.
AFTER READING THE MS IT SEEMS THAT THERE IS ONE MORE IMPORTANT GOAL OFTHE STUDY. YOU LOOKED FOR THE MODEL WHICH FITS THE DATA THE BEST AND TESTTHE VERHULST-PEARL LOGISTIC MODELS. THIS ASPECT OF THE STUDY SHOULD BEBETTER EXPLAINED AND PRESENTED. ANSWER: Thank you very much for the appreciation, it is true. REWRITTEN.
159: The previous data for the exit of motile mites of the predator A. swirskii (Trial 2) (Section2.2) may be better interpreted by the mathematical model used.
THIS SENTENCE NEED BETTER EXPLANATION. “BETTER” than WHAT? WHICH MODEL WERE COMPARED? WHY EXACTLY DID YOU DECIDED TO TEST VERHULST-PEARLLOGISTIC MODELS?
ANSWER: With the data from a previous work [44] we evaluated the fit of these data to different mathematical models: mathematical models: differential equations of the Lotka-Volterra type, simpler models of logistic growth of the Verhulst-Pearls type, Markov chains, etc. We found that the logistic models were the best fit to the data of predatory mite outbreaks from the sachets. Do you consider it necessary to provide more information in this paper?
209: According to the results obtained here, the RH influenced the population dynamics of A. swirskii and C. lactis inside the slow-release sachets along with the number of predators that were released.
CONSIDERING YOUR STATEMENT IN LINES 41-43 THIS STATMENT SEEMS OBVIOUS. COULD YOU EMPHASIZE WHAT EXACTLY NEW (CONCERNING RH AND MITES) WAS DONE IN YOUR STUDY? REWRITTEN.
216-217: The internal C. lactis population did not increase in any humidity regime and decreased gradually, but the decrease was faster at low RH than at medium or high humidity.
COULD YOU PROVIDE ANY BIOLOGICAL EXPLANATION FOR THIS FINDING IN THE DISCUSSION? ACCORDING TO THE FIG.2B IT LOOKS LIKE C.LACTIS REPRODUCE FASTER UNDER MEDIUM RH DURING THE FIRST WEEK BUT LATER… (THIS NEEDS SOME DESCRIPTION/EXPLANATION). ALTHOUGH YOU GIVE SOME EXPLANATION IN THE FURTHER TEXT, IT IS TOO REMOTE FROM THIS 216-217 SENTENCE. DELETED.
231-234: In addition, with this regime of medium RH, the humidity of the sachet remained above 60% until Day 10, which favored better biological parameters of both species and the release of predators (the water content of the substrate was reduced by half at the end of the trial), as will be described in more detail below. 2
“as will be described in more detail below” THIS SENTENCE NEEDS REVISING. DELETED.
440-442: (8) In the microclimate of greenhouses, particularly RH, plant growth of spring-summer crops and at the beginning of the growing season is lower, which may lead to ambient RH conditions that influence the actions of predatory mites, regardless of their release system, but including notably those formulated in slow-release sachets.
THIS IS NOT A CONCLUSION, THIS A IS A HYPOTHESIS. THIS WOULD BE BETTER TO MENTION IN THE DISCUSSION AND DREMOVE FROM CONCLUSIONS DELETED.: (8). It is already included in the Discussion (Lines 341-350).

Reviewer 3 Report

Manuscript is original, well defined and understandable. The topic is compatible with the journal’s aim. The results are very significant and relevant, presented in a well-structured manner. The manuscript’s results are reproducible based on the details given in the methods section. The figures (1-4) and tables (1) are appropriate, they are clearly presented. Conclusions justified and supported by the results, consistent with the evidence and arguments presented. References are listed according to the regulations of the publishers.

Line 3 - I suggest that it be specified taxonomic rank of both mite species (Order: Family) in the title of your manuscript.

Author Response

Reviewer #3
The authors are grateful for the work of Reviewer #3 and for the comments expressed about the MS.
Major comments to the Author:
Manuscript is original, well defined and understandable.
The topic is compatible with the journal’s aim.
The results are very significant and relevant, presented in a well-structured manner. The manuscript’s results are reproducible based on the details given in the methods section. The figures (1-4) and tables (1) are appropriate, they are clearly presented.
Conclusions justified and supported by the results, consistent with the evidence and arguments presented. References are listed according to the regulations of the publishers.
Minor comments:
Line 3 - I suggest that it be specified taxonomic rank of both mite species (Order: Family) in the title of your manuscript. ANSWER: Thank you for the indication, but we consider that the title is long. Including the taxonomic groups would make it even longer. On the other hand, the journal does not request that taxonomic groups be included in the titles.
